# Placental Mitochondrial Function and Dysfunction in Preeclampsia

**DOI:** 10.3390/ijms24044177

**Published:** 2023-02-20

**Authors:** Fahmida Jahan, Goutham Vasam, Alex E. Green, Shannon A. Bainbridge, Keir J. Menzies

**Affiliations:** 1Department of Biochemistry, Microbiology and Immunology, Faculty of Medicine, University of Ottawa, Ottawa, ON K1H 8M5, Canada; 2Interdisciplinary School of Health Sciences, Faculty of Health Sciences, University of Ottawa, Ottawa, ON K1H 8M5, Canada; 3Ottawa Institute of Systems Biology, University of Ottawa, Ottawa, ON K1H 8M5, Canada; 4Department of Cellular and Molecular Medicine, Faculty of Medicine, University of Ottawa, Ottawa, ON K1H 8M5, Canada

**Keywords:** placenta, mitochondria, preeclampsia, disease subclasses, pregnancy, hypertension, reactive oxygen species, therapies

## Abstract

The placenta is a vital organ of pregnancy, regulating adaptation to pregnancy, gestational parent/fetal exchange, and ultimately, fetal development and growth. Not surprisingly, in cases of placental dysfunction—where aspects of placental development or function become compromised—adverse pregnancy outcomes can result. One common placenta-mediated disorder of pregnancy is preeclampsia (PE), a hypertensive disorder of pregnancy with a highly heterogeneous clinical presentation. The wide array of clinical characteristics observed in pregnant individuals and neonates of a PE pregnancy are likely the result of distinct forms of placental pathology underlying the PE diagnosis, explaining why no one common intervention has proven effective in the prevention or treatment of PE. The historical paradigm of placental pathology in PE highlights an important role for utero–placental malperfusion, placental hypoxia and oxidative stress, and a critical role for placental mitochondrial dysfunction in the pathogenesis and progression of the disease. In the current review, the evidence of placental mitochondrial dysfunction in the context of PE will be summarized, highlighting how altered mitochondrial function may be a common feature across distinct PE subtypes. Further, advances in this field of study and therapeutic targeting of mitochondria as a promising intervention for PE will be discussed.

## 1. Introduction

The human placenta serves as a metabolic and endocrine exchange barrier between the gestational parent and fetus, regulating adaptation to pregnancy and fetal growth and development [1,2]. Abnormalities in the development and function of this organ underlie the majority of obstetrical disorders faced today, many of which can be classified under the umbrella term of “placenta-mediated diseases”. One of the most common of these is preeclampsia (PE), a hypertensive disorder of pregnancy that affects ~3–5% of all pregnancies and to date has no effective therapeutic interventions [3]. A large body of research has focused on understanding the placental pathology(/ies) that lead to gestational parent hypertension and end-organ damage. Much of this research has focused on a paradigm of utero–placental malperfusion, placental hypoxia and oxidative stress resulting in shedding of placental debris, and proinflammatory and anti-angiogenic mediators into the circulation of the gestational parent. This initiates a heightened inflammatory response and endothelial dysfunction that precipitate the clinical symptoms observed in the gestational parent [3,4]. 

The primary functions of the placenta are to regulate gestational parent–fetal exchange (gas exchange, nutrients/micronutrient and ion exchange, waste removal); hormone and growth factor production (required for uterine quiescence, gestational parent adaptation to pregnancy, fetal growth and development); and to act as a selective barrier to the fetus (fetal protection from xenobiotics, and enzymatic inactivation of gestational parent hormones, i.e., cortisol) [5]. Each of these functions is highly energy dependent and consequently highly reliant on mitochondria function.

Mitochondrial dysfunction has been widely implicated as a central component to the establishment of placental and gestational parent disease in cases of PE [6]. Mitochondria not only harvest energy to produce adenosine triphosphate (ATP) but also play a vital role in signaling pathways that are essential for growth, development, and homeostasis of any organ. Mitochondrial dysfunction is a central mediator of disease pathophysiology in many organ systems, owing to the numerous processes that can be directly influenced by mitochondrial function, including biosynthetic and bioenergetic pathways, reduction–oxidation (redox) signaling, antioxidant defense, hypoxia adaptations, inflammatory signaling, the mitochondrial unfolded protein response, calcium signaling, and apoptotic signaling [7]. Similarly, when placental mitochondrial functions become dysregulated, placental development and function may be compromised, resulting in detrimental health outcomes for both the developing fetus and the gestational parent [6,8,9,10]. In the current review, literature on the central role of mitochondria in the establishment of placental disease will be summarized, highlighting the possibility that placental mitochondrial dysfunction may be a central and common aspect of pathophysiology across distinct subclasses of PE. Further, recent advancements in mitochondria-targeting therapeutics to treat PE will be discussed. 

## 2. Mitochondria in the Healthy Placenta

Nutrient and energy demand of the placenta change across pregnancy to match and optimize fetal development and growth patterns [11,12,13]. Further, the placenta requires a considerable amount of ATP for its own growth and expansion, for example, during developmental processes that include vascularization, cell differentiation and the formation of function-specific structures within the placenta [14]. Mitochondrial content, structure, and function play an important role in supporting the placenta’s energetic demands during these highly dynamic transitions. A brief overview is provided below and summarized in Table 1, but a more comprehensive discussion on its role in healthy placental development and function can be found in the following reviews [6,15,16,17]. 

### 2.1. Placental Mitochondrial Content, Structure, and Function across Pregnancy

During most of the first trimester of human pregnancy, feto-placental development takes place in a low-oxygen environment [18,19,20], accompanied by low placental mitochondrial content. At this gestational phase, mitochondria found within the chorionic villi structures are relatively large in size, with an oval or circular shape. At the onset of the second trimester, when the gestational parent–fetal circulation is established, placental mitochondrial biogenesis is upregulated, leading to increased mitochondrial content that continues to rise until the end of pregnancy [21]. Mitochondrial structure during this trimester likewise demonstrates dynamic remodeling, with the presence of smaller mitochondria with irregular cristae, likely a function of rapid placental mitochondrial turnover in response to increased oxygen supply [22]. During this gestational phase, placental respiration and ATP production increases; however, the ratio of respiration rates to mitochondrial content demonstrates a slowing within the third trimester of pregnancy, perhaps to maintain respiration capacity for acute energetic insults [21]. 

In addition to meeting the specific temporal energy demands, the placental mitochondria must be capable of maintaining cellular redox homeostasis across pregnancy in the face of a changing oxygen landscape. Mitochondria are a major source of reactive oxygen and nitrogen species (ROS, RNS) generation. They also house several key antioxidant enzymes, required to maintain cellular redox balance [23]. When compared with pre-pregnancy state, an increase in the maternal blood levels of superoxide is required for healthy pregnancy; and this level remains stable throughout the gestation [24]. With the rapid increase in oxygen availability at the transition between first and second trimester, a burst of ROS production is also noted, with elevated ROS production maintained across the remainder of gestation. Importantly, placental ROS generation is considered an important second messenger system to promote development of the placental vasculature [25]. On the other hand, the expression and activity of mitochondrial superoxide dismutase (MnSOD) and glutathione peroxidase (GPX) enzymes could be upregulated in response to increased oxidative stress as an adaptive mechanism under both physiological and pathological conditions [22,26]. In line with this, results from a mouse study showed that placental mitochondria are highly responsive to environmental changes, such as oxygen level, and can modulate its oxidative capacity and substrate preference accordingly [27]. 

Placental mitochondria are also involved in regulating cellular apoptosis. Upon cellular stimuli such as nutrient stress, mitochondrial membrane permeability may increase, leading to swelling and release of proapoptotic proteins to initiate the mitochondria mediated caspase 9 apoptotic cascade [28]. The mitochondrial-driven apoptotic cascade is a central biological process required for normal placental developmental processes, such as gestational parent–fetal immune tolerance, trophoblast differentiation, syncytium formation, trophoblast invasion and remodeling of uterine spiral arteries [29,30]. Thus, aberrant regulation of apoptosis within the placenta at any point in pregnancy can disrupt placental development and function and contribute to the establishment of placental dysfunction and obstetrical disease.

### 2.2. Mitochondrial Content, Structure, and Function in Distinct Placental Cell Types

The human placenta demonstrates considerable anatomical complexity, with heterogenous structures and cellular composition—characteristics required to support the diverse functions of this critical organ of pregnancy. Unsurprisingly, the distinct placental cell populations demonstrate unique oxygen availability and energetic needs, dramatically influencing the mitochondrial content, structure, and function in the distinct populations. As such, in addition to temporal plasticity in the placental mitochondrial landscape, there is also considerable heterogeneity in mitochondrial content, structure and function across the distinct cell populations and structures within the placenta. 

In humans, the syncytiotrophoblast (STB) cellular layer (also referred to as the syncytium) of the chorionic villi, serves as the barrier between the circulations of the gestational parent and the fetus. The syncytium has long been considered the metabolic workhorse of the placenta, due to its direct roles in gestational parent–fetal nutrient/gas exchange and hormone production. However, recent findings suggest that primary human placental villous cytotrophoblasts (CTBs)—cells that terminally differentiate and fuse into the overlying syncytium—have higher mitochondrial content and respiration rates than syncytiotrophoblast (STB) in culture. Mitochondria in CTBs are large and oval shaped with defined cristae, while mitochondria in differentiated STB are smaller and round with undefined cristae, which may reflect cell-specific roles and metabolic properties for these respective mitochondrial pools [31]. Functionally, mitochondria are important for steroid biosynthesis via their role in isoprenoid precursor synthesis and, in particular, mitochondria found in the STBs constitutively produce progesterone to sustain pregnancy. It is thought that mitochondrial size and cristae structural changes occur during CTB to STB differentiation to support improved cholesterol import into the STB mitochondria, supporting efficient production of progesterone [32,33,34,35]. 

A third trophoblast cell population—the extravillious cytotrophoblasts (EVTs)—demonstrate high energy demands associated with their functionality, namely, invading the uterine decidua and remodeling the uterine vasculature. Using the immortalized EVT cell line (HTR-8/SVneo) in vitro, paracrine signaling from placenta-derived mesenchymal stem cells has been shown to increase expression of mitochondrial proteins important for ATP production, such as ATP-binding cassette sub-family B member 10 (ABCB10) and mitochondrial Ca+ uniporter (MCU), the latter of which also acts as a cellular glucose sensor [36,37]. Furthermore, compared with the villous counterparts, the EVTs exhibit increased mitochondrial respiration and produce more ATP [38]. Moreover, activation of mitochondrial quality control process, mitophagy and balancing mitochondrial dynamics is known to be essential in successful EVT invasion and required for proper placentation [36,37]. 

In summary, mitochondria play a central role in ensuring proper development and functioning of the human placenta, supporting the development and growth of the fetus and successful pregnancy outcomes. Conversely, dysregulation of mitochondrial content, structure and/or function can have negative consequences for placentation and pregnancy health, and has been identified as a significant contributor to the pathophysiology of all placenta-mediated obstetric complications, including preeclampsia (PE)—one of the most prevalent and devastating of the placenta-mediated diseases [37].

## 3. Preeclampsia

PE is one of the most common and serious placenta-mediated diseases in humans, contributing to approximately 76,000 gestational parent and 500,000 fetal/neonatal deaths per year worldwide [3,39]. PE is characterized by de novo hypertension after 20 weeks of gestation, accompanied by proteinuria and/or evidence of end-organ damage in the gestational parent [40]. Fetal growth restriction is a common comorbidity, particularly in cases of early onset PE, estimated to occur in ~39% of PE cases [41]. Adverse neonatal health outcomes associated with PE include periventricular or intraventricular hemorrhage or subdural and cerebral hemorrhage, hypoxic–ischemic encephalopathy or periventricular leukomalacia, stillbirth, and perinatal death [40], in many cases the iatrogenic results accompanying medically-indicated preterm delivery. Aside from the immediate health risks associated with a PE diagnosis (i.e., HELLP, eclampsia), individuals who have had PE are seven times more likely to have a recurrence of PE in subsequent pregnancies, particularly with early onset disease [42]. Further, these individuals are at increased risk for cardiovascular disease in later life, with some estimates indicating the average age of first cardiac events as early as 38 years of age [42]. 

The underlying cause of PE is not entirely clear, however placental dysfunction is recognized as a critical component of the pathophysiology. In a healthy pregnancy, uterine spiral arteries are remodeled by the EVT population, modifying them into large-diameter vessels that are no longer responsive to vasopressors in the circulation of the gestational parent; modifications aimed at increasing blood flow into the intervillous space of the placenta to promote fetal growth and development [43]. In the dogmatic two-stage model of PE pathophysiology [44], it is proposed that this vital aspect of placentation becomes compromised, with shallow invasion of the EVTs and insufficient remodeling of the uterine spiral arteries [45,46,47,48]. This poor vascular remodeling results in placental ischemic–reperfusion injury; the high speed, pulsatile blood flow into the placenta causing damage to the delicate villous structures, promoting thrombus formation and compromising adequate gestational parent–fetal exchange of oxygen and nutrients. This injury is supported by findings of STB necrosis, damage and destruction of syncytial microvilli structures, hyperplastic and apoptotic CTBs, and bulged endoplasmic reticulum cisternae and mitochondria [9,49,50]. Damaged trophoblasts are proposed to respond to the structural damage and oxidative stress through the upregulation and release of pro-inflammatory cytokines and anti-angiogenic factors, along with extracellular vesicles, apoptotic debris and cell-free fetal DNA into the systemic circulation of the gestational parent [51,52,53]. In the second stage of the PE pathophysiology model, these placenta-derived factors illicit a heightened systemic immune response and endothelial dysfunction in the gestational parent, precipitating in the clinical manifestation of the disease [54,55]. While there is certainly robust evidence to support this model of PE pathophysiology, it is important to note that not all cases of PE demonstrate evidence of insufficient invasion and uterine spiral artery remodeling, nor do all PE placentas demonstrate structural or molecular evidence of ischemia–reperfusion type injury [4]. 

## 4. Subclasses of Disease in Preeclampsia

PE is a heterogeneous disorder that demonstrates considerable variability in the timing of disease onset, presentation and severity of symptoms, gestational parent and fetal health outcomes, and pathological evidence of placental disease—indicative of multiple disease subtypes with divergent underlying etiology and pathophysiology. In this vein, the scientific community has embarked on several lines of inquiry attempting to identify and characterize distinct subtypes of PE [4,56,57,58,59,60,61]. Discovery-driven approaches using robust omics datasets are particularly useful when trying to understand complex diseases with heterogeneous presentation and are proving useful in gaining insight into the presence of distinct PE subclasses. Using clustering approaches, groups of PE patients who share commonality in (epi-)genome [62], transcriptome [4] or proteome [59] profiles in relevant tissues (i.e., blood, placenta) have been identified. Our group has contributed to this body of work, completing multi-scale profiling of a robust sample of PE patients that spanned the clinical spectrum of disease [4,63,64]. Through integration of clinical, placental transcriptome and histopathology profiles, our group uncovered at least three clinically relevant PE subclasses (Table 2) with divergent forms of placental disease, or lack thereof, which are highly supported by the growing body of research in this domain (see [65] for a detailed review on this topic by the Global Pregnancy Collaboration).

### 4.1. Canonical PE Subclass

The largest PE subclass characterized by our previous work demonstrates a phenotype which tightly aligns with the traditional framework of PE pathophysiology. These patients have small placentas with lesions of maternal vascular malperfusion (MVM) and gene expression profiles consistent with hypoxia–reperfusion injury—including upregulation of hypoxia-inducible factors and anti-angiogenic molecules commonly associated with PE. Clinically, these patients have an earlier onset and more severe form of gestational parent disease and are more likely to experience preterm delivery (<34 weeks) [4,63].

### 4.2. Immunological PE Subclass

Heightened systemic inflammation has long been identified as a key component of PE pathophysiology [66,67,68], and, more recently, it is becoming clear that an immune-driven subclass of PE exists [69,70,71]. This subclass of PE patients demonstrates evidence of chronic inflammatory insults at the utero–placenta interface, with heightened immune cell infiltration, a propensity for gestational parent–fetal HLA-discordance, heightened placental expression of pro-inflammatory cytokines and chemokines, and placental lesions consistent with chronic inflammation and gestational parent–fetal interface disturbances [4]. 

### 4.3. Gestational Parent-Driven PE Subclass

Despite being widely identified as a placenta-mediated disease, historical reports have long described a handful of PE cases with little evidence of placental disease [4,63,72]—a PE subclass driven in large part by underlying and predisposing factors in the gestational parent, which compromise appropriate physiological adaptation to pregnancy. This subclass of PE patients demonstrates normally grown placentas with gene expression profiles that mimic healthy tissues and little histological evidence of pathology. Fetuses in this subclass demonstrate normal growth trajectories and gestational parent symptoms are relatively mild with a later onset (>37 weeks gestation) [4,63]. Collectively, this phenotype suggests minimal placental involvement, but rather predisposing gestational parental factors as the primary drivers of PE pathophysiology.

While there is convincing evidence supporting distinct mechanistic pathways driving placental and/or gestational parent disease across PE subclasses, there is likewise convincing evidence to suggest that mitochondrial dysfunction may be a central point of convergence within the pathophysiology of PE across all subclasses of patients. 

## 5. Placental Mitochondrial Dysfunction in Preeclampsia

Compromised placental mitochondrial biogenesis, structure, and functionality contribute to placental dysfunction in PE. Below, the existing evidence for placental mitochondrial dysfunction as a key component of PE pathophysiology—as a singular clinical diagnosis—is reviewed (Table 1).

### 5.1. Mitochondrial Content

The evidence related to mitochondrial content in cases of PE is not clear cut. Decreased placental mitochondrial DNA content and citrate synthase protein activity have been described in cases of PE, compared with healthy term controls, coupled to a downregulation of genes that regulate mitochondrial biogenesis, such as PGC1-α, NRF1 and Tfam [73]. It should be noted, however, that when PE placentas are compared against gestational age-matched controls, the reported difference in citrate synthase levels are no longer apparent, and an increase in mitochondrial complex II and III protein levels are observed [74]. In a subsequent prospective case control study, placental mtDNA copy number was in fact found to be elevated in cases of preterm/severe PE, compared with both term PE and healthy term controls, possibly indicative of a compensatory upregulation of mitochondrial biogenesis in the face of severe placental disease [75]. Systemically, lower cell-free (membrane bound and non-membrane bound) plasma mtDNA content has been measured in the circulation of individuals with PE in the third trimester, compared with gestational age-matched healthy controls [76]. However, others have reported elevated whole blood, plasma and serum mtDNA content in cases of PE compared with health controls, with the mtDNA content elevations significantly associated with female fetal sex according to one of these studies [77,78,79]. On its own, the evidence surrounding placental mitochondrial content in the context of PE does not appear to be highly informative—however the evidence suggests the timing of disease onset, severity of disease, gestational sampling time and fetal sex may all influence the necessary interpretations of mitochondrial content measurements. 

### 5.2. Mitochondrial Structure and Dynamics

To maintain mitochondrial homeostasis in the face of environmental stress, mitochondria go through dynamic morphological changes via two processes, fusion and fission. The fusion of separate mitochondria is driven by outer mitochondrial membrane proteins mitofusins 1 and 2 (Mfn1 and 2), and the inner membrane protein optic atrophy 1 (OPA1) [80]. The benefits of fusion have been linked to the diffusion of metabolites, protein and mtDNA across the network of mitochondria, enabling mitochondria to produce energy more efficiently. Fission, on the other hand, permits the separation of damaged portions of mitochondria for degradation via mitochondrial autophagy (i.e., mitophagy), and is mediated by in large part by mitochondrial fission 1 (FIS1) and dynamin-1-like (DNM1L) proteins [81]. 

Evidence for dysregulated mitochondrial dynamics (fission and fusion) has been collected from placenta tissues of pregnancies impacted by PE. Term PE placenta tissues demonstrate decreased mRNA expression of the fusion protein Mfn2, coupled to decreased ATP content, when compared with gestational age-matched controls [82]. While not explicitly examined in that study, similar decreases in Mfn2 expression have been associated with mitochondrial fragmentation in both human and rodent models of pulmonary arterial hypertension [83]. Structurally, placental mitochondria from cases of severe/early onset PE demonstrate swelling, impaired cristae and fragmentation when visualized with electron microscopy [84,85,86,87]. Fission processes are also likely impacted in early onset PE cases, with reported measurements of increased DNM1L expression relative to term controls [73]. Interestingly, cultured first trimester immortalized human placenta cells (TEV-1) likewise exhibit decreased expression of Mfn2, decreased mitochondrial membrane potential, decreased ATP production, mitochondrial fragmentation, and mitophagy, when exposed to hypoxic insults in vitro [82]—similar to what is proposed to occur within the context of PE pathophysiology. Damaged mitochondrial clearance via mitophagy is described in cases of PE, particularly those with severe/early onset disease [88]. This finding is paralleled by an increase in placental expression of mitophagy-related proteins, such as PTEN-induced putative kinase 1 (PINK1), Parkin, B-cell lymphoma-2 (BCL-2) interacting protein 3 (BNIP3) and BNIP3-like (BNIP3L), when compared with preterm controls [73,88].

### 5.3. Mitochondrial Function- Respiration and ATP Generation

The mitochondria primarily act as an ATP generation hub through their ability to perform oxidative respiration. Altered placental mitochondrial respiration is widely reported in cases of PE, however the direction of this functional change varies across studies. It should be noted that performing mitochondrial functionality assays using fresh samples (immediately after collection) is ideal; however fresh tissue collection can pose a difficulty in multi-site clinical studies or when performing retrospective studies, especially in pregnancy research. Unfortunately, freezing of samples can lead to damage of mitochondrial membranes, uncoupling ETC activity and loss of cytochrome c. Thus, historical results must be interpreted with caution when frozen tissue was used. To mitigate such issues, newly reported techniques that restore electron transfer components lost during freeze/thaw and correct for membrane permeabilization, have demonstrated considerable success, preserving 90–95% of the maximal respiratory capacity in frozen samples, and may serve as tremendously useful for future research in this area (see reference [89] for method).

In cryopreserved placental samples collected from cases of early onset PE (onset at <34 weeks), a considerable decrease in mitochondrial respiration is described, when compared with healthy term controls [87]. Supporting these findings, enzymes critical to the glycolytic pathway have been found to be upregulated in placentas from PE pregnancies, indicative of suppressed mitochondrial respiration [73]. Muralimanoharan et al. more specifically described decreased mitochondrial complex-I and complex-III activity in isolated mitochondria from frozen PE term placentas, when compared with gestational age-matched controls, functional findings that were coupled to decreased expression of respiratory complex proteins [85].

In contrast, others have described an increase in placental mitochondrial respiration activity in cases of PE. For example, Vishnyakova et al. [86] demonstrated a modest increase in complex-I mediated respiration, with no change in complex-II mediated respiration, when studying freshly isolated mitochondria from placentas of early onset PE patients compared with healthy term controls. It should be noted, however, this same study found no measurable differences in mitochondrial activity in late onset PE samples, when compared with gestational age-matched controls—highlighting the importance of considering the temporal changes of mitochondrial content and activity across pregnancy when interpreting this data. However, others have reported an increase in complex-I and complex-I + II mediated respiration in term PE permeabilized placenta tissue that occurred in conjunction with increases in the expression of respiratory protein complexes, when gestational age was appropriately matched in the healthy control group [74]. Importantly though, the reserve respiratory capacity—the amount of extra ATP that can be produced by OXPHOS with an energetic demand—was reduced in these same PE samples [74], suggesting these mitochondria were likely functioning close to their maximal capacity. These findings may also imply that PE cases that do reach term may do so because of a unique capacity to induce compensatory increases in mitochondrial function. 

There are several possible reasons for the high degree of study finding discrepancies on this topic. One possible interpretation may be that those placentas capable of initiating a compensatory increase in mitochondrial respiratory function at times of cellular stress, can limit the severity of placental disease and allow the pregnancy to progress to term. Conversely, placentas unable to initiate this response succumb to a more severe form of placental disease and dysfunction, initiating an earlier onset and more severe form of PE. Alternatively, these discrepancies may in fact be further evidence of distinct subclasses of PE, with divergent underlying pathophysiology and placental disease. It should be noted that in all studies described, no efforts to identify distinct PE subclasses were attempted, aside from the clinical distinction of early vs. late onset disease. 

### 5.4. Mitochondrial Function—Redox Homeostasis and Apoptosis

Mitochondrial complexes I and III are among the major sites of cellular ROS generation [90], particularly under hypoxic conditions [91,92]. Oxidative stress, due to excessive ROS generation, is a well described hallmark of PE pathophysiology [9,93]. For instance, elevated levels of oxidative stress markers such as 8-oxoDG, an indicator of DNA damage, and lipid peroxides in the placenta and maternal plasma in PE cases have been reported [94,95]. Elevated levels of ROS, such as H_2_O_2_, have been measured in both the placentas and circulating serum of PE patients compared with healthy controls, however it is not entirely clear where the systemic ROS is originally generated [96]. Evidence of increased mitochondrial lipid peroxidation in PE placentas corresponds to high levels of lipid peroxide in the circulation of the gestational parent [95,97,98,99]. Moreover, higher levels of dichlorofluorescin (DCF) oxidants, reactive nitric oxide metabolites such as NO_2_^−^ and NO_3_^−^, and protein carbonyl, have been observed in the placental tissue mitochondria from preterm/severe PE patients compared with term healthy controls, implying a major contribution of mitochondrial ROS in the oxidative stress observed in PE [100].

Coupled to increased ROS generation, there is also evidence of adaptive changes in antioxidant capacity in the context of PE. For example, mitochondrial superoxide dismutase (MnSOD) enzymatic activity is decreased in preterm/severe PE placentas compared with gestational age-matched controls, while its mRNA levels have been shown to be upregulated [74,101]. Some studies suggest that there is an overall increase in antioxidant capacity of PE placenta through activation of other antioxidant enzymes present in cytoplasm and mitochondria such as glutathione peroxidase (GPx) [73,74]. This suggests that an adaptive response is required in order to maintain placental function in the oxidative stress condition. 

Intracellular stress such as oxidative damage can activate the mitochondria mediated caspase-9 apoptotic pathway [102]. Intracellular stress induces activation of proapoptotic proteins Bax and Bak that inactivate anti-apoptotic Bcl-2 and Bcl-xL proteins, leading to opening of mitochondrial membrane permeability pores [103]. PE placentas show increased presence of apoptotic syncytial knots with evidence of oxidative DNA damage [104,105,106]. Holland et al. showed that the ratio of BAX/BCL2 protein expression was increased in preterm/severe PE placentas compared with gestational age-matched controls [74]. On the other hand, in term PE placentas, the ratio of BAX/BCL2 expression was relative to term controls while the levels of antioxidant enzymes increased [74]. These findings suggest that increased antioxidant activity in PE placentas may keep oxidative stress at bay and, in turn, reduce cellular apoptosis and facilitate reaching term pregnancy.

## 6. Mitochondrial Dysfunction—A Point of Convergence across Preeclampsia Subclasses

To date, most research focused on understanding placental mitochondrial function and dysfunction in PE combines all cases together or uses the clinical classification scheme of “early-onset” vs. “late-onset”. While this foundational work has been tremendously helpful in gaining an appreciation for the central role of mitochondrial dysfunction in PE, it is important to note that even within these clinically distinct groups of patients, there remains a high degree of heterogeneity in: (1) the types of placental disease and damage noted at the time of pathological review; (2) circulating concentrations of placenta-derived biomarkers of disease (i.e., sFLT, sENG); (3) placental gene expression profiles; and (4) impact on fetal growth profiles. Collectively, this data would indicate that even within these clinically distinct populations, their very likely exists different underlying etiology and pathophysiology, and detailed profiling work conducted by our group [4,61,63,64] and others [65], certainly supports this conclusion. Hence, to gain a true appreciation for the role of mitochondrial dysfunction in the establishment and progression of PE, it is important to consider the underlying disease processes at play.

### 6.1. Canonical PE Subclass—Ischemia-Reperfusion Induced Mitochondrial Dysfunction

The historical dogma of PE pathophysiology includes defective EVT invasion of the decidua and impaired spiral artery remodeling, leading to hypoxia–reoxygenation events and oxidative damage in the placenta [107,108]. The canonical PE subclass previously described by our group demonstrated gene expression, histopathology and clinical findings aligned to this disease model [4]. Data collected in rodent models of PE established by surgical reductions in uterine blood flow—the RUPP model—demonstrated a 30% increase in placental H_2_O_2_ production, compared with sham treated controls [109], highly indicative of placental mitochondrial redox dysfunction. Likewise, in primary human trophoblast cells, hypoxic culture conditions (2–5% O_2_) increased the production of mitochondrial ROS, triggering HIF1-α-mediated expression of the anti-angiogenic PE marker sFLT-1—a circulating vascular endothelial growth factor signaling inhibitor [110,111]. Increase in sFLT-1 not only causes systemic endothelial dysfunction in the gestational parent but also induces further mitochondrial dysfunction and oxidative stress in placenta [112]. A recent study of placentas from early onset ‘canonical PE’ with evidence of ischemia–reperfusion injury, demonstrated that activation of mitochondrial unfolded protein response (UPRmt) led to a decrease in the expression of mitochondrial quality control protease Caseinolytic peptidase P (CLPP) [87]. In vitro, knockdown of CLPP in the BeWo trophoblast cell line led to increased mitochondrial fission and reduced respiration, mimicking findings of canonical human PE placentas. Recently, another study demonstrated that chronic hypoxia in ovine pregnancy induced ER and cytoplasmic UPR, however, activation of mitochondrial UPR was not investigated [113].

### 6.2. Immunological PE Subclass—Inflammation Induced Mitochondrial Dysfunction

Preeclampsia has long been associated with a pro-inflammatory state, however a clearer identification of PE cases that demonstrate profound immune dysregulation at the utero–placenta interface, in the absence of traditional hallmarks of ischemia–reperfusion injury, has helped to clarify the existence of this unique subclass of PE patients [4,74,114,115]. Due to the understudied nature of this PE subclass in general, our current state of knowledge on the mechanisms through which chronic inflammation at this interface establishes placental dysfunction, and more specifically, what role placental mitochondria may play in the placental pathophysiology within this subclass, are not currently well understood. However, a wealth of literature has identified mitochondrial dysfunction as a common feature of many chronic inflammatory diseases in non-pregnant populations, as well as within the aging process [116,117,118]. Inflammatory mediators have been shown to alter the activities of respiratory complexes, ATP production, mitochondrial membrane potential and its morphology [118]. Proinflammatory cytokines such as TNF-α have been shown to be an inducer of both cytoplasmic and mitochondrial ROS generation [119,120,121,122]. A recent study suggests that inflammation induced by TNF-α causes reverse electron flow through mitochondrial complex I and, thus, stimulates mitochondrial ROS production [122]. 

Increased levels of TNF-α have been measured in placentas of PE cases [114,123,124], and profiling work carried out by our group demonstrated a very high placental gene expression of both TNF-α and TNF-α receptors uniquely within the ‘immune-driven PE subclass’ samples [4]. Interestingly, daily infusions of TNF-α (50 ng/day) in a pregnant rat model was capable of eliciting PE clinical features, including hypertension and decreases in both glomerular filtration rate and renal plasma flow [125]. In vitro treatment of first trimester placental explants with TNF-α (10 ng/mL) compromised key aspects of placental function, including migratory and invasive properties [126]. In the JEG3 trophoblast cell line, TNF-α treatment (0.2–5.0 ng/mL) prevented cell integration into endothelial networks and tube formation [127]. While there is certainly strong evidence to suggest that TNF-α, in addition to other pro-inflammatory signaling molecules, can initiate placental dysfunction and contribute to adverse pregnancy outcomes, the specific role of placental mitochondria, or mitochondrial dysfunction, in the context of chronic inflammation in pregnancy is not yet as clear. In vitro, TNF-α treatment in primary human trophoblasts, isolated from female offspring originating from overweight or obese pregnant individuals, caused decreased expression of mitochondrial complex-I subunit NDUFA4 and iron–sulfur cluster assembly enzyme (ISCU) and inhibited mitochondrial respiration [128]. In an LPS-induced mouse model of intrauterine inflammation, placental transcriptomic analysis demonstrated dysregulated expression of upstream regulators of mitochondrial function. Further, metabolomics analysis showed alterations in TCA cycle and increase in acylcarnitine levels, known to induce mitochondrial dysfunction and oxidative stress [129,130]. Collectively, this body of work that is still in its infancy, certainly suggests that inflammation alone can initiate mitochondrial dysfunction in placental tissues, and as such would lend support to the hypothesis that mitochondrial dysfunction may in fact be a common point of convergence in the establishment of placental disease across multiple PE subclasses. 

### 6.3. Gestational Parent Driven PE—Predisposing Factors and Mitochondrial Dysfunction

Contrary to the traditional dogma of PE pathophysiology, in which placental dysfunction is central to disease establishment, there have long been reports of PE cases with minimal evidence of placental disease [4,131,132]. Our group has confirmed the existence of this patient population through detailed clinical and placental profiling, concluding this group of patients likely arrive at a PE diagnosis (hypertension and end organ damage) as a result of poor gestational parent adaptation to pregnancy, rather than placental dysfunction [4]. In healthy pregnancies, the process of placentation and normal placental function results in the release of some level of placenta-mediated factors into the gestational parent’s circulation, including: antiangiogenic factors (i.e., sFLT-1), cytokines and chemokines, microparticles/microvesicles, exosomes, apoptotic bodies and cell-free fetal DNA (to name a few) [133,134,135,136]. In a healthy pregnancy, the gestational parent responds to these signaling factors in a way that promotes appropriate adaptation to pregnancy. However, predisposing risk factors (i.e., diabetes, obesity, advanced age, kidney disease, systemic lupus, antiphospholipid syndrome, chronic hypertension) may prevent the appropriate adaptive response in the gestational parent and instead may trigger systemic pathophysiology, including a heightened immune response, endothelial damage and dysfunction and the development of hypertension. As discussed in the previous section, a pro-inflammatory environment may be a potent trigger for mitochondrial dysfunction. Further, the dysregulated role of mitochondria within the context of endothelial dysfunction and hypertension has already been widely described, even in the context of pregnancy (see reviews [17,137,138] for more details). As such, while there may be a subclass of PE patients in which placental mitochondrial dysfunction may not be a major contributing factor to disease, systemic mitochondrial dysfunction in the gestational parent is still likely to be of major concern—and this is likely true of all subclasses of PE described. In fact, elevated mutational loads of circulating cell-free (ccf)-mtDNA have been identified in blood (buffy coat) collected from third trimester patients with PE and gestational hypertension alike. As ccf-mtDNA is often regarded as a marker of systemic inflammation, this result suggests that dysregulated mitochondria from gestational parent origin is a likely contributor to PE pathophysiology [139]. In vitro studies have shown that treatment of human umbilical vascular endothelial cells (HUVECs) with serum of PE patients or exogenous placenta-mediated factors (sFLT-1 or angiotensin II type 1 autoantibodies), leads to decreased mitochondrial respiratory capacity, increased superoxide formation and induces a glycolytic phenotype [79,140,141]. Thus, with or without the direct contribution of placental disease, systemic mitochondrial dysfunction within the gestational parent is likely to be another common point of convergence of pathophysiology across all PE subclasses. 

Collectively, the current literature supports a role for mitochondrial dysfunction, both within the placenta and within gestational parent tissues, in the pathophysiology of PE. The underlying etiology of disease may in fact be different across PE subclasses, but we propose that dysregulation of mitochondrial biogenesis and function may be a common feature to all cases of PE (Figure 1), and as such, serve as an important therapeutic target in the identification of effective treatments for this common and debilitating disorder of pregnancy.

## 7. Potential for Mitochondrial Targeting Therapies in Preeclampsia

Currently, there are no highly effective treatments for PE. Antihypertensive drugs are used to manage clinical symptoms in the gestational parent, but these treatments are not aimed at targeting the source of disease and as such these treatments neither serve to prolong the pregnancy nor ameliorate adverse impacts of PE on fetal growth profiles [142]. As there is sufficient evidence to implicate mitochondrial dysfunction in the establishment and/or progression of PE, consideration should be given to how therapeutic strategies aimed at improving mitochondrial function may be applied in cases of PE [6,8]. In this section, some of the most recent advancements on this topic will be discussed (Figure 2).

### 7.1. Resveratrol

Several studies have shown that mitochondrial complex-I inhibitors may provide benefits in disease conditions by reducing superoxide generation through complex-I. Resveratrol (3,4,5-trihydroxy-trans-stilbene) has been described as one of them. It is a plant-based polyphenol found in some fruits such as grapes, berries, and peanuts. It has antioxidant, anti-inflammatory and anti-microbial properties. Studies also suggest that it enhances mitochondrial homeostasis by activating the master regulator of mitochondrial biogenesis, peroxisome proliferator-activated receptor gamma coactivator 1-alpha (PGC-1ɑ) (see the following review for more information [143]). Notably, it reduced blood pressure in hypertensive rodent models [144]. Resveratrol administration also improved uterine artery blood flow velocity and increased fetal weight in a pregnant mouse model of uterine blood flow defect [145]. Similarly, resveratrol has shown some beneficial effects in various in vitro PE models. In one study, first trimester HTR-8/SVneo trophoblast cell lines were treated with angiotensin II, TNF-α, protein kinase C (PKC) activator, phorbol-12-myristate-13-acetate (PMA) for 24 h. These treatments resulted in increased expression of sFLT1—a major contributor to gestational parent endothelial dysfunction in PE. However, co-treatment with resveratrol significantly reduced the expression of this potent antiangiogenic factor and did not demonstrate any cytotoxic effect on the cells [146]. Interestingly, in a randomized control trial carried out with 400 patients diagnosed with PE, the use of resveratrol as an adjuvant treatment alongside oral nifedipine (an anti-hypertensive drug commonly prescribed in severe PE) was shown to further improve both systolic and diastolic blood pressure compared with nifedipine treatment alone. With the resveratrol-combination therapy, time required for controlling blood pressure was reduced and time before developing new hypertension was delayed; thus, indicating a beneficial effect of resveratrol [147]. Interpretation of these results must be taken with caution, however, as there was no resveratrol alone in the study arm, rather the patients all received nifedipine as well. Thus, it is not clear based on these study results alone that resveratrol may serve as an effective solo therapy.

### 7.2. Metformin

Metformin is an effective treatment for type 2 diabetes that lowers blood glucose levels and reduces insulin resistance. It has anti-inflammatory and antioxidant effects [148,149]. Metformin has been shown to increase the expression of mitochondrial sirtuin 3, an NAD^+^ dependent deacetylase, that regulates oxidative phosphorylation and activates mitochondrial antioxidant MnSOD [149,150]. It also activates adenosine 3′,5′-monophosphate (AMP) protein kinase (AMPK), a major regulator of mitochondrial fission, mitophagy and biogenesis [151,152]. Metformin has been used during pregnancy to improve pregnancy outcomes in individuals with obesity, gestational diabetes, type 2 diabetes and polycystic ovary syndrome. Interestingly, metformin has been shown to reduce the production of the anti-angiogenic factors sFLT1 and sENG (endoglin) in both villous cytotrophoblasts and endothelial cells in culture [153]. Moreover, third trimester placental villous explants collected from PE patients treated with metformin in culture demonstrated a decrease in the secretion of sFLT1 and sENG, and co-treatment with succinate blocked this effect, indicating metformin-mediated regulation of sFLT1 and sENG expression is likely mediated via mitochondrial function [153]. In a PE mouse model induced via a high fat diet, metformin administration was also shown to reduce blood pressure and increase circulating levels of vascular endothelial growth factor (VEGF) [154]. The data in humans collected to date, however, is not as clear. A number of clinical trials, observational cohort studies and subsequent meta-analyses have been carried out to assess the ability of metformin treatment to reduce the establishment of PE in individuals at high risk, with mixed results [155,156,157,158,159]. Collectively, the data suggest that metformin may prove to be beneficial for a specific subset of patients, specifically those that have polycystic ovary syndrome (PCOS) [156,157,158]. Moreover, metformin treatment in the context of gestational diabetes has been associated with low birth weights, indicating a potential fetal risk with this form of treatment [6,160]. 

### 7.3. MitoQ and MitoTEMPO

MitoQ is a synthetic ubiquinone that is intracellularly converted to ubiquinol by the enzymatic activity of mitochondrial complex II. Ubiquinol serves to neutralize free radicals and can be continuously recycled to maintain a potent antioxidant presence in the mitochondria. Similarly, MitoTEMPO, which can specifically be targeted to the mitochondria via a bound lipophilic cation, contains piperidine nitroxide (Tempo) that mimics superoxide dismutase and catalyzes the dismutation of the superoxide radical into molecular oxygen and hydrogen peroxide. These mitochondria-targeted antioxidants have shown therapeutic promise in the RUPP rodent model of PE, where they have been shown to significantly reduce the maternal blood pressure and increase both placental and fetal weights. [109]. In another study, MitoTEMPO improved placental mtDNA stability and mitochondrial dynamics in a PE-like rat model [161]. In cell culture, it reduced mitochondrial ROS production in HUVECs treated with plasma from PE patients. Both MitoQ and MitoTEMPO have been tested in humans and found to be tolerable with no adverse effect [162,163,164,165]. MitoQ has been reported to reduce liver inflammation in patients with chronic hepatitis C virus (HCV) infection [163]. On the other hand, MitoTEMPO has been shown to improve cutaneous vascular conductance (CVC) in patients with chronic kidney disease [164]. However, caution should be taken when considering therapeutic application of such antioxidants in pregnancy as one of the rodent studies indicated that MitoQ administration early in pregnancy could hinder normal placenta development by reducing physiological ROS levels [166].

### 7.4. Nicotinamide (NAM)

Nicotinamide or NAM is a derivative of vitamin B3 and has antioxidant and anti-inflammatory properties [167,168]. NAM is a known cellular nicotinamide adenine dinucleotide (NAD^+^) donor and NAD^+^ level is critical for maintenance of OXPHOS function, mitochondrial biogenesis, and homeostasis [169]. NAD^+^ is also important for DNA damage response, inflammation, and oxidative stress pathways as it functions as a cofactor for certain enzymes such as PARPs and sirtuins [170]. The therapeutic potential of NAM treatment has been investigated using three different rodent models of PE: RUPP, sFLT1 overexpression and ASB4 (ankiryn-repeat-and-SOCS-box-containing-protein) knockout mice. In all three models, NAM administration (500 mg/kg per day) was shown to decrease blood pressure, proteinuria and prevent glomerular endotheliosis in the mothers. Further, NAM treatment increased fetal and placental weight, prolonged pregnancy duration and increased the number of live births. NAM also decreased HIF1-alpha expression; and increased the levels of NAM, NAD^+^ and ATP in fetal brains [171]. As NAM is a NAD^+^ donor, it is likely that the observed beneficial effects were through improving mitochondrial function at various maternal or fetal tissue levels, including placenta. Thus, more research needs to be performed to understand the mechanism. Moreover, there is a safety concern regarding the use of NAM at high dose during pregnancy. Studies performed in rats showed that NAM administration at 500 mg–4 g/kg/day dose was associated with liver toxicity [172,173]. Maternal NAM intake led to DNA hypomethylation in rat placenta and fetal liver [174]. Indeed, in humans, NAM intake at 300 mg/day was shown to elevate plasma levels of homocysteine, an indicator for methyl donor consumption due to NAM methylation [175]. However, there are additional NAD^+^ donors that can be examined for their efficacy in improving mitochondrial and placental function, and ultimately pregnancy outcomes, such as nicotinamide riboside (NR), nicotinamide mononucleotide (NMN), dihydronicotinamide riboside (NRH)—each of which may demonstrate a more appropriate safety profile for use in pregnancy. 

## 8. Future Perspective

Findings from animal and human studies indicate that mitochondrial dysfunction is a central component to the establishment of both placental disease and gestational parent disease in PE. The degree of mitochondrial dysfunction and inherent ability to adapt to cellular stress signals may dictate the severity of placental or gestational parent tissue damage and, ultimately, pregnancy outcome. Novel therapeutic interventions aimed at targeting such common disease features will likely prove to be highly effective at the population level regardless of their subclass in a clinical setting. However, there are still many hurdles to overcome in this domain. Using animal models and human studies, future research should aim to: (1) identify specific aspects of mitochondrial dysfunction that are common across PE and should be the focus of therapies moving forward; (2) determine the critical window when mitochondrial dysfunction occurs during PE pathogenesis in order to maximize therapeutic benefits, and (3) vigorously test the safety profiles of these treatments, as any novel treatments must not interfere with placental or fetal development. 

## Figures and Tables

**Figure 1 ijms-24-04177-f001:**
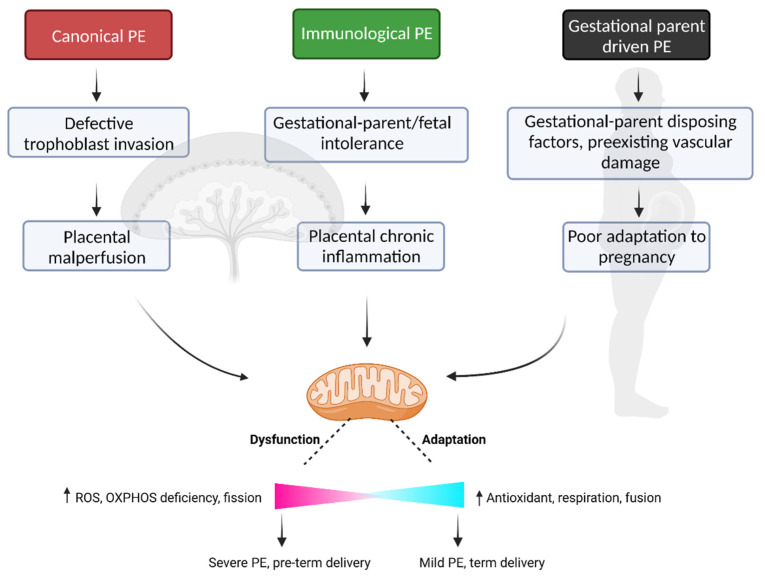
Mitochondrial dysfunction as a common feature across all PE subclasses. PE disease driven by either placental malperfusion, chronic inflammation at the utero–placental interface and/or poor gestational parent adaptation, likely all demonstrate altered mitochondrial respiration, increased ROS production and mitochondrial fragmentation within the placenta and across numerous gestational parent tissues. The inherent ability of the mitochondria to adapt during pregnancy to various pathological insults may in part dictate the clinical outcome of the pregnancy, with high degrees of mitochondrial adaptation associated with a milder clinical presentation of the disease, prolonged gestation, and improved fetal growth profiles. The “↑” arrow indicates an increase. Figure was created with Biorender.com, accessed on 24 January 2023.

**Figure 2 ijms-24-04177-f002:**
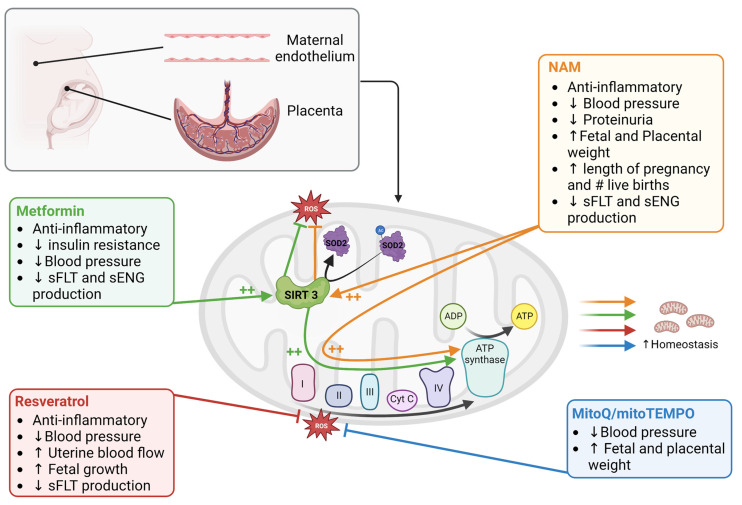
Mitochondria targeting treatments in preeclampsia. Among the drugs and nutraceuticals tested in PE studies, the most promising ones that target mitochondria are metformin, resveratrol, nicotinamide (NAM), and mitoQ/mitoTEMPO. Metformin, resveratrol, and NAM function on multiple pathways that improve mitochondrial biogenesis, function, quality control (by mitophagy), and reduce ROS; thereby establishing mitochondrial homeostasis under pathological conditions. MitoQ/mitoTEMPO act by reducing mitochondrial ROS which eventually improve mitochondrial homeostasis. The symbol ++ indicates increased activation, # indicates number, ↑ indicates increase and ↓ indicates decrease. Figure was created with Biorender.com, accessed on 13 February 2023.

**Table 1 ijms-24-04177-t001:** Overarching trends of placental mitochondria content, structure/dynamics, and function in healthy pregnancies (early and late) and PE pregnancies (early onset vs. late onset).

		Healthy Pregnancy				PE	
		Early		Late		Early-Onset/Severe	Late-Onset/Mild
**Content**		**⇓**	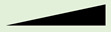		**⇑**	**⇑⇓**	**⇑=**
**Structure and dynamics**		Large Oval or circular		Smaller Irregular cristae		Swelling Impaired cristae Fragmentation	Similar to healthy More fusion?
**Function**	**Respiration and ATP generation**	**⇓**	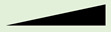		**⇑**	**⇑⇓**	**⇑=**
	**ROS**	**⇓**	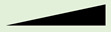		**⇑**	**⇑**	**⇑=**
	**Antioxidant** **capacity**	**⇓**	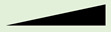		**⇑**	**⇑⇓**	**⇑**

In healthy pregnancy, comparison was between early and late gestation. In PE-complicated pregnancy, comparison was with healthy term or preterm pregnancy. The symbol “**=**“ was used when the finding was similar to healthy control. The symbol ↑ indicates increase and ↓ indicates decrease.

**Table 2 ijms-24-04177-t002:** Overview of clinical features and mechanistic pathways involved in the pathophysiology of distinct PE subclasses.

	Canonical PE	Immunological PE	Gestational Parent Driven PE
**Onset and severity of** **disease**	Early onset (many <34 weeks) and ↑ preterm delivery More severe symptoms (↑ blood pressure, ↑ proteinuria, HELLP) Low birth weight newborns (SGA and FGR)	Early–late onset (34–37 weeks) and near-term delivery ↑↑ growth restricted infants (FGR)	Late onset (>37 weeks) and term delivery Less severe symptoms Appropriate birth weight for gestational age newborns
**Pathogenesis**	Impaired spiral artery remodeling Placental ischemia–reperfusion injury	Gestational parent/fetal immune incompatibility Pro-inflammatory response	No evidence of placental disease Likely driven by gestational parent factors
**Genes expressed in placenta**	Hypoxia and oxidative stress responses such as egl-9 family hypoxia–inducible factor 1 (EGLN1), BCL family of proteins, etc. Anti-angiogenic molecules (FLT, ENG) Hormone signaling (INHA)	Immune and inflammatory responses (TNF-ɑ, IFN-Ɣ, CXCL-10)	Unchanged from healthy controls
**Placenta pathology**	Smaller placenta Maternal vascular malperfusion lesions (infarcts, distal villous hypoplasia, excessive syncytial knots)	Chronic inflammation lesions (i.e., villitis of unknown etiology, massive perivillous fibrin deposition) Infiltration of immune cells	Normal weight placenta No evidence of placental disease
**Potential risk factors**	Advanced age of gestational parent Pre-existing chronic hypertension and diabetes Genetic heredity	Type A blood Short duration paternal antigen exposure Non-Caucasian ethnicity	Nulliparity Advanced age of gestational parent High pre-pregnancy BMI Pre-existing chronic hypertension

The “↑” arrow indicates an increase.

## Data Availability

No new data created.

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
