# Peer review of "Placental Mitochondrial Function and Dysfunction in Preeclampsia"

_ijms, 2023, doi:10.3390/ijms24044177_

Round 1

Reviewer 1 Report

Placental Mitochondrial Function and Dysfunction in Preeclampsia

                In this review the authors describe mitochondrial structure and function in a healthy placenta and the contribution of mitochondria towards normal placental function and fetal growth. The authors go on to describe the dysfunction  that takes place in human preeclamptic placentas and in animal models of preeclampsia. The authors finish by reviewing several therapeutics that could improve mitochondrial function and thereby improve placental function and the preeclamptic syndrome. The review covers an important area in preeclampsia research and putting a light on potential therapeutics is an important contribution to the field. The authors’ conclusions and recommendations for future work are sound and will move the field forward in a positive direction/

1.       Use maternal or mother rather than gestational-parent.

2.       Rather than grouping PE into the three subclasses “Canonical PE, Immunological PE, and Gestational-Parent Driven PE” it would be more appropriate to use early-onset, late-onset, and postpartum PE. Examining the literature to show that mitochondrial dysfunction bridges these three subgroups would be more impactful and relevant to the current field.

3.       It might be useful to include a table showing the changes in mitochondrial structure and function in preeclampsia.

4.       Including the disclaimer on mitochondrial respiration results from frozen tissues earlier might be helping in understanding why there are divergent results. A clearer description of new techniques to mitigate this issue and support the use of these techniques would be relevant.

5.       The link between Metformin and mitochondria function is not clear and should be strengthened.

6.       A figure showing the molecular mechanisms that the reviewed therapeutics take to improve PE symptoms would be helpful.

7.       The spacing changes after the first table.

8.       Define GPX at first use on line 91.

9.       Line 421 needs a paragraph break and indention.

Reviewer 2 Report

Jahan et al. reviewed the mitochondria structure and function in healthy placentas and preeclamptic placentas. They specifically focused on the potential roles of mitochondria in PE pathogenesis, and the potnetial therapeutic strategies for PE that target mitochondria. It is a well written article, and gives the readers new view about the roles of mitochondria in PE development.

Major comments:

1.       Although the authors have described the structure and function of mitochondria in health placenta, the roles of mitochondria in regulation of trophoblast function and placental development should be paid much more attention.  

2.       Several important articles should be cited. For instance, “Placental mitochondria adapt developmentally and in response to hypoxia to support fetal growth” (PMID: 30655345). About the therapeutic targeting michondria studies, Yang et al (PMID: 32228063) and Long et al (PMID: 36009224) have described very interesting data about the roles of mitoQ and MitoTEMPO in animal models of PE.
